# Role and modulation of various spinal pathways for human upper limb control in different gravity conditions

**Alice Bruel** [1] *, **Lina Bacha** [1], **Emma Boehly** [1], **Constance De Trogoff** [1], **Luca Represa** [1], **Gregoire Courtine** [2], **Auke Ijspeert** [1]

**1** Biorobotics Laboratory, EPFL, Lausanne, Switzerland, **2** NeuroRestore, EPFL, Lausanne, Switzerland

* alice.bruel@epfl.ch

**Data Availability Statement:** For reproducibility and comparative purposes, the source code is available on github at https://github.com/AliceBrue/sc_modulation.git.

## Abstract

Humans can perform movements in various physical environments and positions (corresponding to different experienced gravity), requiring the interaction of the musculoskeletal system, the neural system and the external environment. The neural system is itself comprised of several interactive components, from the brain mainly conducting motor planning, to the spinal cord (SC) implementing its own motor control centres through sensory reflexes. Nevertheless, it remains unclear whether similar movements in various environmental dynamics necessitate adapting modulation at the brain level, correcting modulation at the spinal level, or both. Here, we addressed this question by focusing on upper limb motor control in various gravity conditions (magnitudes and directions) and using neuromusculoskeletal simulation tools. We integrated supraspinal sinusoidal commands with a modular SC model controlling a musculoskeletal model to reproduce various recorded arm trajectories (kinematics and EMGs) in different contexts. We first studied the role of various spinal pathways (such as stretch reflexes) in movement smoothness and robustness against perturbation. Then, we optimised the supraspinal sinusoidal commands without and with a fixed SC model including stretch reflexes to reproduce a target trajectory in various gravity conditions. Inversely, we fixed the supraspinal commands and optimised the spinal synaptic strengths in the different environments. In the first optimisation context, the presence of SC resulted in easier optimisation of the supraspinal commands (faster convergence, better performance). The main supraspinal commands modulation was found in the flexor sinusoid's amplitude, resp. frequency, to adapt to different gravity magnitudes, resp. directions. In the second optimisation context, the modulation of the spinal synaptic strengths also remarkably reproduced the target trajectory for the mild gravity changes. We highlighted that both strategies of modulation of the supraspinal commands or spinal stretch pathways can be used to control movements in different gravity environments. Our results thus support that the SC can assist gravity compensation.

**Funding:** This work was supported by the the European Union Human Brain Project Specific Grant Agreement 3 (H2020-RIA. 945539), awarded to AI. The funders had no role in study design, data collection and analysis, decision to publish, or preparation of the manuscript.

**Competing interests:** The authors have declared that no competing interests exist.

## Summary

Human movement relies on coordination between the brain, spinal cord, and musculoskeletal system to adapt to various environments and gravity conditions. While the brain plans movements, the spinal cord contributes through reflexes and local control. It remains unclear how these systems adjust, whether primarily at the brain, spinal level, or both. This study explored upper limb control in different gravity conditions using computational models. Simulations integrated brain-driven commands and spinal reflexes to examine their roles in maintaining adaptive movement. The research revealed that both brain and spinal cord play complementary roles in adapting to environmental changes. The spinal cord helps compensate for gravitational differences, reducing the demands on brain-level control. These findings offer insights into the interplay between neural and musculoskeletal systems, advancing our understanding of how humans adapt movement to dynamic physical contexts.

## 1 Introduction

Humans can perform movements in various physical environments and various positions (corresponding to different experienced gravity). This is not a trivial achievement as it requires adjusting motor commands to reflect complex changes in body dynamics. The interaction of the musculoskeletal system, the neural system and the external environment is necessary to generate such movements [1]. The neural system is itself comprised of several interactive components, from the brain mainly conducting motor planning [2], to the spinal cord (SC) implementing its own motor control centres through pattern generators and sensory reflexes. Nevertheless, it remains unclear whether similar movements in various environmental dynamics necessitate replanning modulation at the brain level, correcting modulation at the spinal level, or both.

One major hypothesis in control theory postulates the existence of internal model encoded in the brain [3]. The brain would thus solves inverse problems mapping the desired trajectories into motor commands by representations of the sensorimotor system and the environment, allowing for the prediction and compensation of changes in the environment. This hypothesis is supported by manipulandum studies in which the arm performs a reaching task while interacting with a new mechanical environment imposed by a force field. With practice, hand trajectories converged to a path very similar to that observed in free space showing that the CNS compensates for the new dynamics. Moreover, after force field removal, mirror aftereffects support the idea of an update of the internal model.

The SC also supports a major role in motor control. The SC indeed integrates descending motor signals from the brain with its own sensory reflex-based mechanisms to regulate muscle activation, and transmit sensory signals back to the brain. The SC can notably generate rhythmic locomotion movements on its own [4, 5]. Sensory feedback are evoked by muscle spindles and Golgi tendon organs (GTO) that are respectively stretch and tension sensitive. These information are then relayed to the motoneurons through spinal circuits enabling fast reflex responses, including the stretch velocity reflex, static stretch reflex, Golgi tendon reflex, and reciprocal inhibition between antagonist muscles [6]. These pathways can also be modulated by descending signals during movements, especially between gait stance and swing phases [7, 8] and during arm movements [9–11]. Despite their ancient evolutionary origins, about 500 million years ago in the first vertebrates [12], the role of spinal circuits in upper limb

control remains unclear. Some experiments highlighted their role in stability and handling perturbations [13, 14], and the presence of activity dependent plasticity mechanisms in the SC in rats, primates and humans [15]. Furthermore, spinal reflex properties adaptation have been reported in hypo and hyper gravity conditions [16], suggesting a role of spinal pathways in gravity compensation. The equilibrium point (EP) hypothesis also states that the CNS can control desired movements by gradually specifying the threshold of the spinal stretch reflexes, and thus shifting equilibrium positions resulting from sets of antagonist muscle forces [17, 18]. In the present study, we address the question of the role of various spinal pathways in upper limb motor control, robustness against perturbation and gravity compensation.

Neuromechanical modeling and simulation serve as powerful tools for studying such complex dynamical mechanisms by integrating body dynamics and physical interactions with the external environment [19]. Previous computational models have investigated various aspects of spinal pathways in upper limb control, from regulatory spinal circuits [20, 21] to reflex modulation and learning at the SC level [22, 23]. Several studies focused on the role of SC in handling perturbation [24–26], arguing for hierarchical levels of feedback and the important contribution of GTO feedbacks. Bruel et al. [27] also showed the role of SC in handling perturbation and facilitating cerebellar motor learning. Other models reproduced the EP hypothesis and captured both the qualitative features and the quantitative kinematic details of human measured movements [28]. Nevertheless, these models present strong limitations including the absence of complex descending signals, or a focus on specific subsets of spinal reflexes. To our knowledge, no model has studied spinal control mechanisms involved in gravity compensation, with different gravity magnitudes and orientations.

To study the role of spinal pathways in upper limb motor control, we implemented a modular spinal cord model integrating supraspinal sinusoidal commands and muscular sensory feedback. The resulting spinal outputs were used to actuate a musculoskeletal upper limb model to reproduce various recorded arm trajectories (kinematics and EMGs). While Bruel et al. [27] focused on how a simple SC interacts with the cerebellum to support motor learning and adapt to perturbations, the current paper aims at exploring a broader range of spinal circuits and their modulation in different contexts. We studied different scenarios evaluating the role of various spinal pathways (such as stretch reflexes) in upper limb motor control, robustness against perturbation and gravity compensation. For this last point, we optimised on one hand, the supraspinal commands, and on the other hand, the spinal synaptic strengths in various gravity conditions (magnitudes and directions) to study supraspinal versus spinal modulation to adapt to different environments. We showed that both strategies of modulation of the supraspinal commands or spinal stretch pathways can be used to control movements in different gravity environments. We optimised motor control parameters within a combined control scheme of supraspinal and SC systems to adapt the model's behaviour. However, as SC reflexes tend to bring the musculoskeletal system back to its resting position (both physiologically and by model definition), it is not intuitive that modulating SC reflexes would enable gravity compensation. To our knowledge, it is the first model studying and revealing spinal control mechanisms involved in gravity compensation.

## 2 Results

We implemented a modular spinal cord model receiving supraspinal sinusoidal commands and muscular sensory feedback to control a musculoskeletal upper limb model as depicted on Fig 1A. We first optimised the supraspinal sinusoidal commands to reproduce various recorded arm trajectories (kinematics and EMGs). The trajectory recordings involved a participant performing repetitive planar elbow flexion-extension and circular movements in the

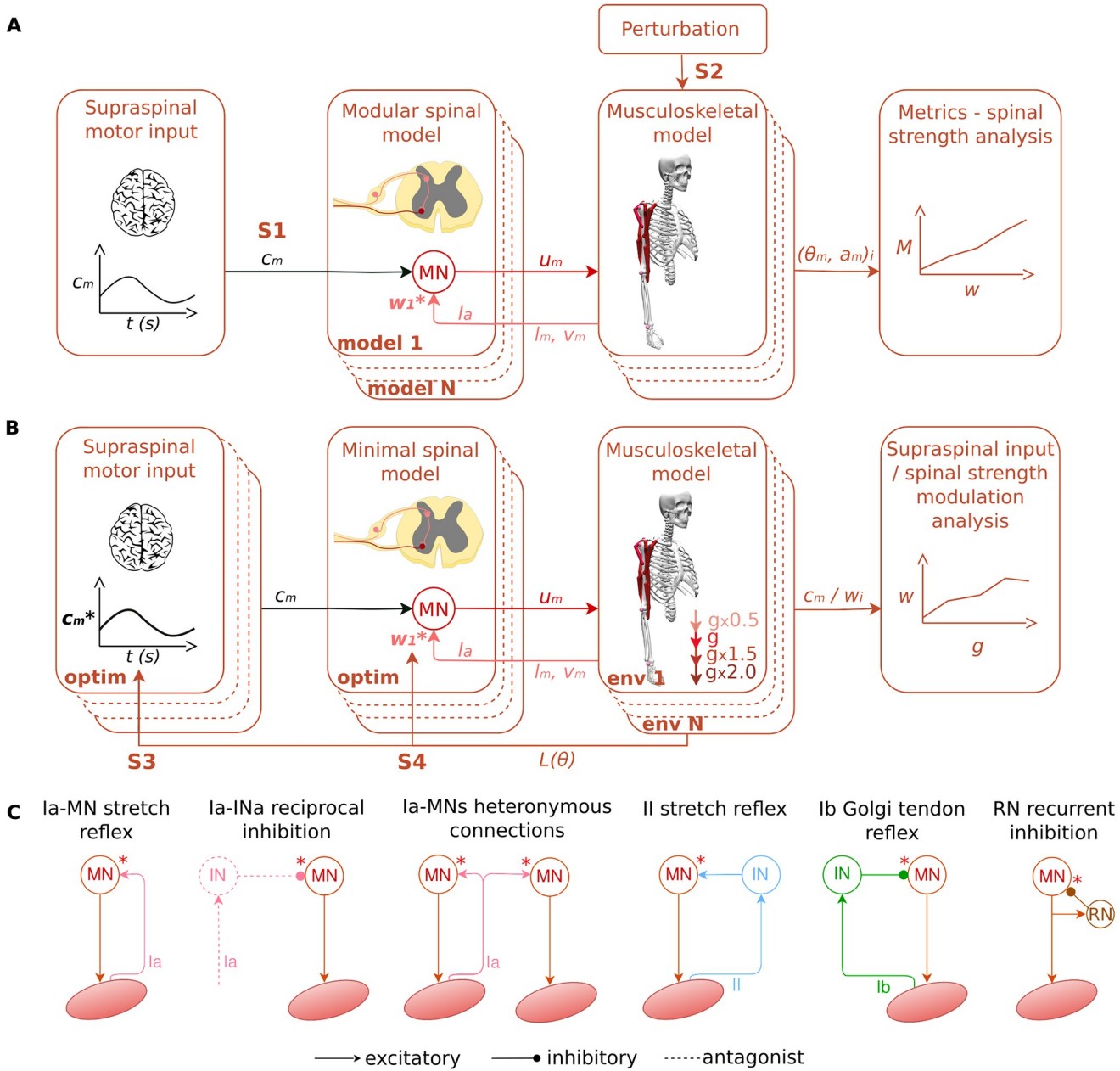

**Fig 1. Modular spinal model in various scenarios.** A) The modular spinal model received supraspinal commands and muscle sensory feedback (length, $l_m$, velocity, $\dot{l}_m$ and force $F_m$) from the musculoskeletal model from OpenSim [29]. Here the stretch reflex is represented as example. The spinal model then generated the final muscle excitation signals ($u_m$) actuating the musculoskeletal model. In the first scenario (S1), supraspinal commands reproducing recorded trajectories were sent. In the second scenario (S2) an additional force perturbation was applied to the hand during the movement. Various spinal pathways with increasing spinal synaptic strengths (model 1 to N) were then evaluated in terms of movement smoothness and robustness against perturbation. B) In the third and fourth scenarios (S3 and S4), a "minimal SC" model comprising Ia-MN stretch reflex and Ia-INa reciprocal inhibition was considered. The supraspinal commands and spinal synaptic strengths were then optimised to reproduce a trajectory in various gravity conditions (magnitudes and directions, env 1 to N) to study supraspinal versus spinal modulation. C) Modeled spinal pathways: Ia-MN stretch reflex, Ia-INa reciprocal inhibition, Ia-MNs heteronymous stretch reflex, II stretch reflex, Ib Golgi tendon reflex and RN recurrent inhibition. The red stars indicate the connection strength that we explored in this study (the others were fixed).

vertical plane in standing position at various self-selected speeds and range of motion. These target trajectories are discrete like reaching and return tasks, and we considered one cycle of sinusoidal command per synergistic muscle group to simplify the optimisation problem, while representing the reaching movement and the return to the initial position. Then, we evaluated various spinal pathways (Ia stretch reflex, reciprocal Ia inhibition, Ia heteronymous stretch reflex, II static stretch reflex, Ib autogenic inhibition and Renshaw cell recurrent inhibition detailed in the Methods section) with increasing synaptic strength to study their effect on motor control in terms of kinematics and muscle recruitment (scenario 1, S1). An additional force perturbation was then applied to the hand during the movement to study the role of spinal pathways in robustness against perturbation (scenario 2, S2). Finally, we considered a "minimal SC" model including the pathways that showed the more notable effect in the previous simulations. We optimised on one hand, the supraspinal commands (scenario 3, S3), and on the other hand, the spinal synaptic strengths (scenario 4, S4) in various gravity conditions (magnitudes and directions) to study supraspinal versus spinal modulation to adapt to different environments (Fig 1B).

The following sections present first the behaviour of the integrated model for several target trajectories, and then the results of the 4 scenarios.

## 2.1 Behaviour of the integrated model for several target trajectories

We first optimised the supraspinal sinusoidal commands for the two scenarios, without and with a "minimal SC", to reproduce a baseline recorded elbow flexion-extension trajectory of 2.8 seconds long and 78˚ range of motion (ROM) and a baseline circular trajectory of 1.3 seconds long and 40˚ of ROM. The shoulder flexion-extension is also unrestricted, so that we consider a double joint control. This way we could check the behaviour of our model for various speeds and ROMs. Our "minimal SC" comprised the Ia-MN stretch reflex and the Ia-INa reciprocal inhibition, this choice is motivated in the next section. We used Covariance Matrix Analysis Evolution Strategy (CMA-ES) optimisation [43]. We repeated each optimisation three times to check the solution space and reproducibility. The resulting loss over optimisation iterations, the final flexor and extensor supraspinal commands, and the resulting shoulder and elbow trajectories compared to the target for the flexion-extension trajectory are depicted on Fig 2A and 2B. The final loss were found to be smaller with SC and the final supraspinal commands were similar over repetitions.

We also compared the resulting muscle activation time windows with recorded EMG activation time windows on Fig 2C. The overlap between simulated and EMG activation time windows was found to be better with SC for both flexor and extensor muscles, especially for extensor muscles.

Our model resulted in similar trends for the circular trajectory. The optimisation with our "minimal SC" is highlighted on Fig 2D.

## 2.2 Scenario 1: Role of various spinal pathways in voluntary movements

We studied the role of six spinal pathways by evaluating their effect on motor control with increasing synaptic strength (scenario 1, S1 on Fig 1A). The six spinal pathways of interest are sketched on Fig 1C. We used the supraspinal commands optimised without SC represented on Fig 2B. We evaluated each spinal pathway individually to facilitate the cause-effect relationship interpretation. For each pathway, we increased the synaptic strength of the last connection to the motoneurons from 0 to 1 (for multi-synaptic pathways, the other synaptic strengths were fixed to 1 as sketched on Fig 1C and detailed in the Methods section). We then quantified the resulting root mean square error (RMSE) to the target, the movement smoothness and the

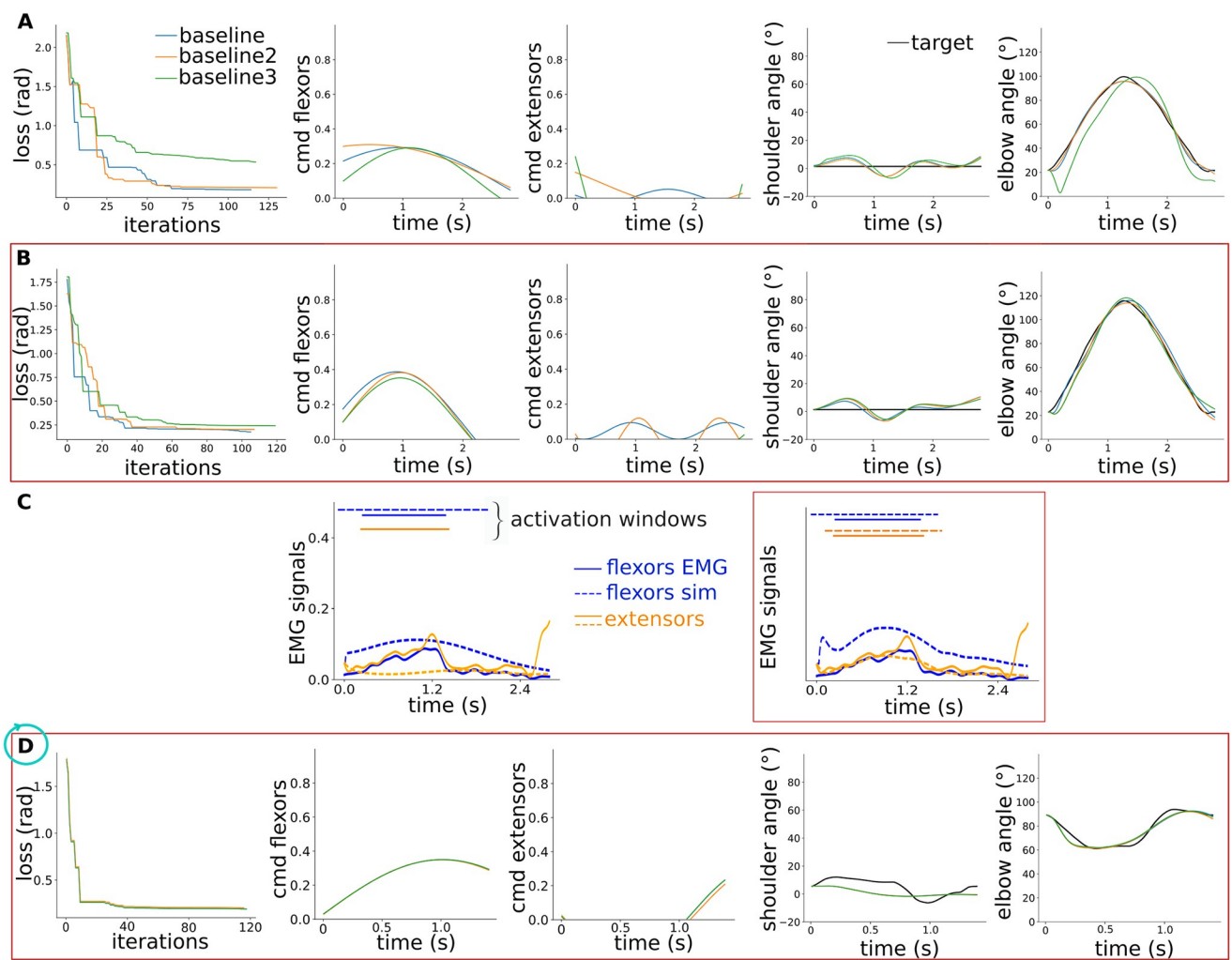

**Fig 2. Behaviour of the integrated model for various target trajectories.** A) CMA optimisation of the supraspinal sinusoidal commands for the flexion-extension trajectory without SC: from left to right the resulting loss over optimisation iterations, the final flexor and extensor supraspinal commands (*cmd*), and the resulting shoulder and elbow trajectories compared to the target for three optimisation runs. **B)** Same with our "minimal SC", highlighted with a red frame. **C)** Resulting muscle activation: activation time windows are compared with recorded EMG activation time windows for flexor and extensor muscles, for both scenarios. **D)** Optimisation of the supraspinal sinusoidal commands for the circular trajectory with our "minimal SC".

overlap between the simulated and recorded EMG activation time windows. Fig 3A shows the shoulder and elbow trajectories for the Ia-MN stretch reflex with increasing synaptic strength. Fig 3B shows the flexor muscle activation for the maximal Ia-MN synaptic strength and the activation time windows comparison with recorded EMG. The resulting metrics for the six spinal pathways with increasing synaptic strength are depicted on Fig 3C. The Ia-MN stretch reflex and Ia-MNs heteronymous connections similarly smoothen the movement (higher elbow speed arc length, SAL with larger synaptic strength). These two pathways do not limit the ROM (small RMSE), in contrast to the II stretch reflex. The Ia-INa reciprocal inhibition resulted in the smallest RMSE. In terms of muscle recruitment, the Ia-MN stretch reflex leads to more physiological patterns (higher EMG overlap) by inducing extensor activation as highlighted on Fig 3B compared to Fig 2C. The other pathways just slightly shaped the flexor activation signal and resulted thus in similar EMG overlap.

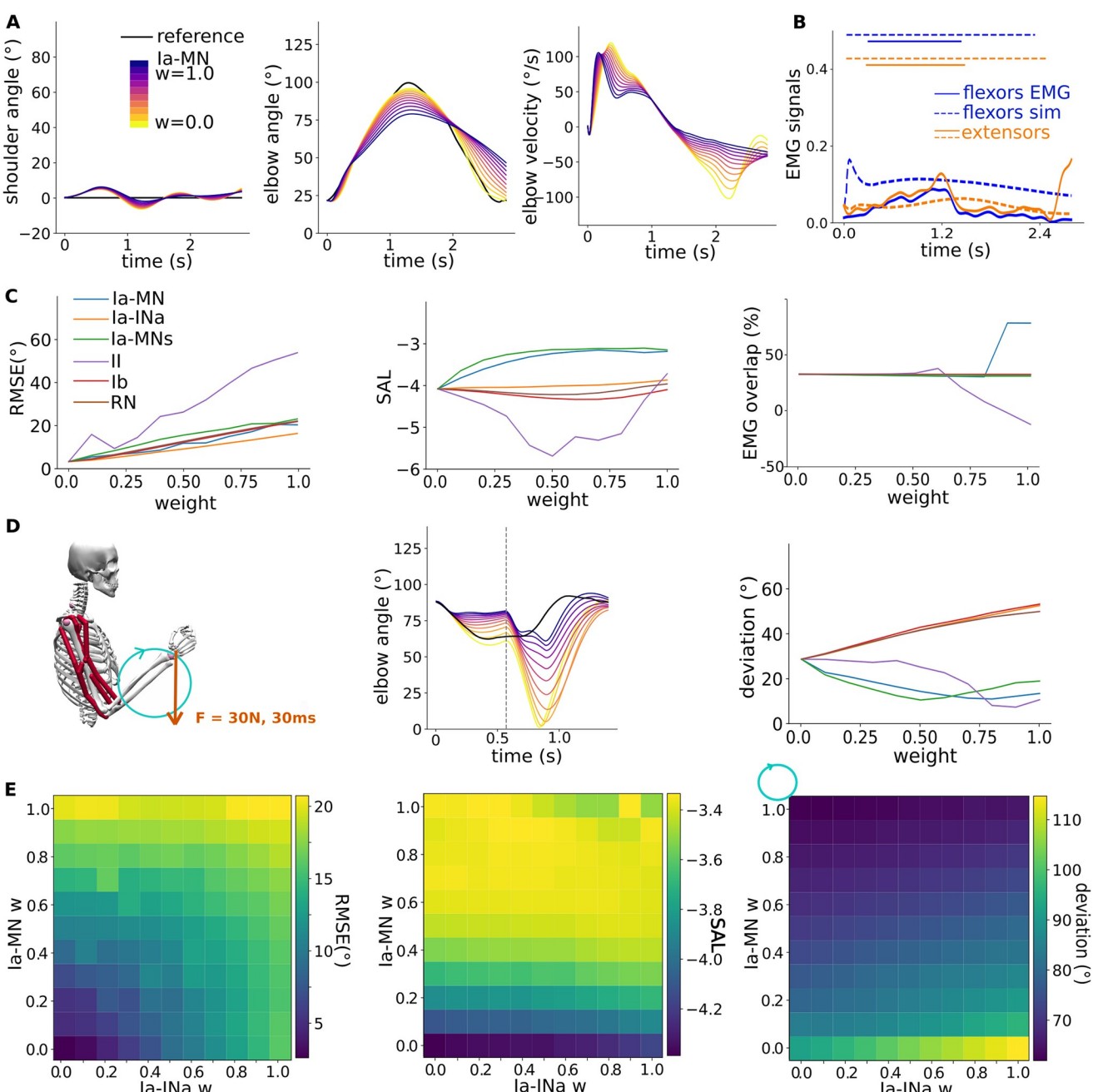

**Fig 3. Role of various spinal pathways in voluntary movements.** A) Shoulder and elbow kinematics for the flexion-extension trajectory with the Ia-MN stretch reflex with increasing synaptic strength. **B)** Resulting flexor and extensor muscle activation for the maximal Ia-MN synaptic strength and the activation time windows comparison with recorded EMG. **C)** Metrics for the six spinal pathways with increasing synaptic strength: root mean square error (RMSE) to the target trajectory, elbow speed arc length (SAL) as smoothness metrics, and EMG time window overlap. **D)** Perturbation study for the circular trajectory: an additional force perturbation was applied to the hand during the movement. The resulting elbow kinematic is showed for Ia-MN pathway, and the deviation to the initial trajectory is plotted for the six pathways with increasing synaptic strength. **E)** Ia-MN—Ia-INa pair analysis: the two pathways are combined to study their interaction.

**Table 1. First-order indices of the sensitivity analysis run on the RMSE and SAL metrics in blue and red respectively.**

|  | *Ia−MN* | *Ia−IN* | *Ia−MNs* | *II* | *Ib* | *RN* |
|---|---|---|---|---|---|---|
| *RMSE* | 0.03 | 0.01 | 0.26 | 0.31 | 0.12 | 0.11 |
| *SAL* | 0.12 | 0.0 | 0.48 | 0.13 | 0.01 | 0.02 |

To assess more precisely the effect of our 6 spinal reflexes, we also performed a sensitivity analysis on the RMSE and SAL metrics. The total-order indices were slightly larger than the first-order indices, indicating mainly first-order interactions. Table 1 summarises the resulting first-order indices. Ia-MNs heteronymous pathways and II stretch reflex have a strong effect on both the RMSE and smoothness metric, while the Ia-MN stretch reflex has a strong effect on the smoothness metric only.

## 2.3 Scenario 2: Role of various spinal pathways in perturbed environment

To study the role of spinal pathways in robustness against perturbation, we then applied an additional force perturbation to the hand during the movement. A 30N force was applied for 30 ms, acting upward during the flexion-extension trajectory and downward during the circular trajectory, when the arm is flexed in both cases as sketched on Fig 3D (scenario 2, S2 on Fig 1B). The resulting elbow kinematics for the Ia-MN pathway show a reduced deviation to the initial trajectory with increasing synaptic strength. We computed the average deviation of the elbow position to the initial trajectory. The Ia-MN stretch reflex, Ia-MNs heteronymous connections and II stretch reflex resulted in decreasing deviation with increasing synaptic strength.

To assess more precisely the effect of our 6 spinal reflexes, we also performed a sensitivity analysis. The total-order indices were slightly larger than the first-order indices, indicating mainly first-order interactions. Table 2 summarises the resulting first-order indices and highlights the strong effect of the Ia-MN stretch reflex and Ia-MNs heteronymous pathways.

We finalised this study by checking the interaction of the Ia-MN stretch reflex and Ia-INa reciprocal inhibition. We chose these two pathways as they implement different reflexes and showed the more notable effects in voluntary and perturbed conditions, while not limiting the ROM. This pair analysis on Fig 3E reveals that strong Ia-MN and Ia-INa pathways together keep the properties of each individual strong pathway; movement smoothness (high SAL), robustness against perturbation (small deviation), while not increasing the RMSE to the target. We thus defined our "minimal SC" including the Ia-MN stretch reflex and Ia-INa reciprocal inhibition with maximal synaptic strength.

## 2.4 Scenario 3: Modulation of the supraspinal commands in various gravity environments

We then studied the modulation of the supraspinal commands without and with our "minimal SC" model in various gravity conditions (magnitudes and directions) as depicted on Fig 4A (scenario 3, S3 on Fig 1B). A gravity direction rotation of 90° thus correspond to the lying

**Table 2. First-order indices of the sensitivity analysis run on the deviation metric.**

|  | *Ia−MN* | *Ia−IN* | *Ia−MNs* | *II* | *Ib* | *RN* |
|---|---|---|---|---|---|---|
| *deviation* | 0.12 | 0.01 | 0.73 | 0.04 | 0.02 | 0.0 |

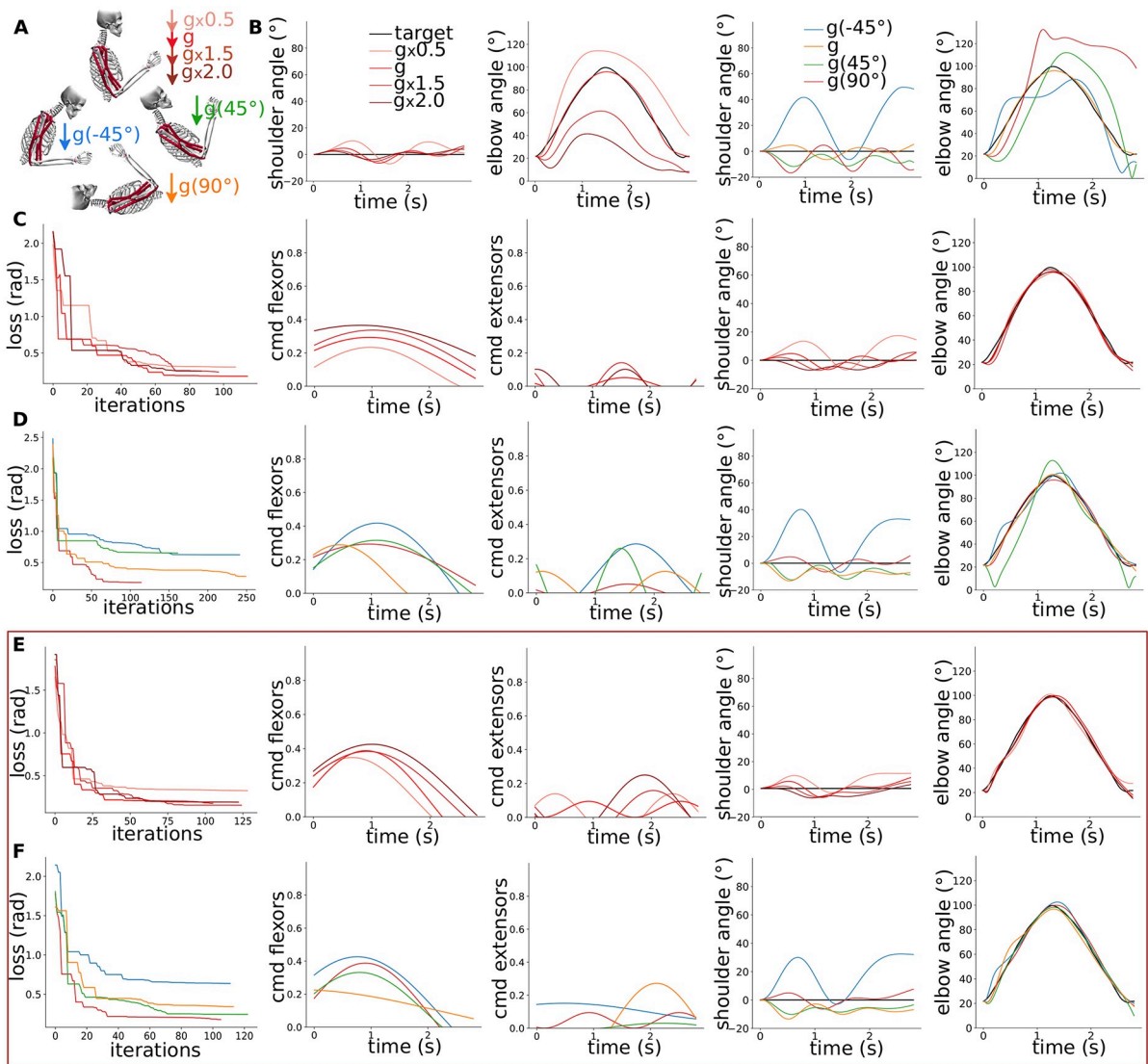

**Fig 4. Modulation of the supraspinal commands in various gravity conditions.** A) Sketch of the 7 gravity environment applied in this study; 4 different magnitudes and 4 different directions. **B)** Resulting flexion-extension trajectories without optimisation; namely with the supraspinal commands from the scenario with "minimal SC" in normal gravity represented on Fig 2B. **C)** Optimisation of the supraspinal commands in various gravity magnitudes without SC: from left to right the resulting loss over optimisation iterations, the final flexor and extensor supraspinal commands (*cmd*), and the resulting shoulder and elbow trajectories compared to the target. **D)** Same in various gravity directions. **E, F)** Same for the scenario with our "minimal SC", highlighted with a red frame.

position. Fig 4B first shows the resulting flexion-extension trajectories without optimisation; namely with the supraspinal commands from the scenario with "minimal SC" in normal gravity represented on Fig 2B. The optimisation of the supraspinal commands without and with our "minimal SC" could reproduce the target trajectory in the various gravity conditions as shown on Fig 4C, 4D, 4E and 4F respectively. The presence of SC resulted in easier optimisation of the supraspinal commands (faster convergence and better performance). The main supraspinal commands modulation was found in the flexor sinusoid's amplitude, resp. frequency, to adapt to different gravity magnitudes, resp. directions.

Similar observations were made for the circular trajectory.

## 2.5 Scenario 4: Modulation of the spinal pathways in various gravity environments

Finally, we studied the modulation of the spinal pathways in the same gravity conditions as above (scenario 4, S4 on Fig 1B). Fig 5A first reveals three optimisations of the spinal synaptic strengths for the flexion-extension baseline in normal gravity condition. We considered the supraspinal commands from the scenario with our "minimal SC" corresponding to Fig 2B. The final synaptic strengths end up to be similar over repetitions especially for the flexor connections. Fig 5B and 5C then show the results of the optimisation of the synaptic strengths in various gravity magnitudes and directions. The modulation of the spinal synaptic strengths also remarkably reproduced the target trajectory for the mild gravity changes. The optimisation in the 2 extreme conditions with 2 times the gravity magnitude and with a gravity direction rotation of -45° could indeed not converge to the target trajectory. Nevertheless, the optimisation results showed two main trends: synaptic strengths were overall larger for all flexor connections and lower for all extensor connections with higher gravity magnitudes. Such modulations indeed facilitate the elbow flexion-extension movement. The synaptic strengths for the different gravity direction conditions did not show that clear trend.

The optimisation of spinal synaptic strengths also successfully reproduced the target circular trajectory, even for the mild and more severe gravity changes. This trajectory indeed exhibit a smaller ROM that facilitates the severe gravity magnitude context, and a symmetry to gravity direction that facilitates the gravity direction context.

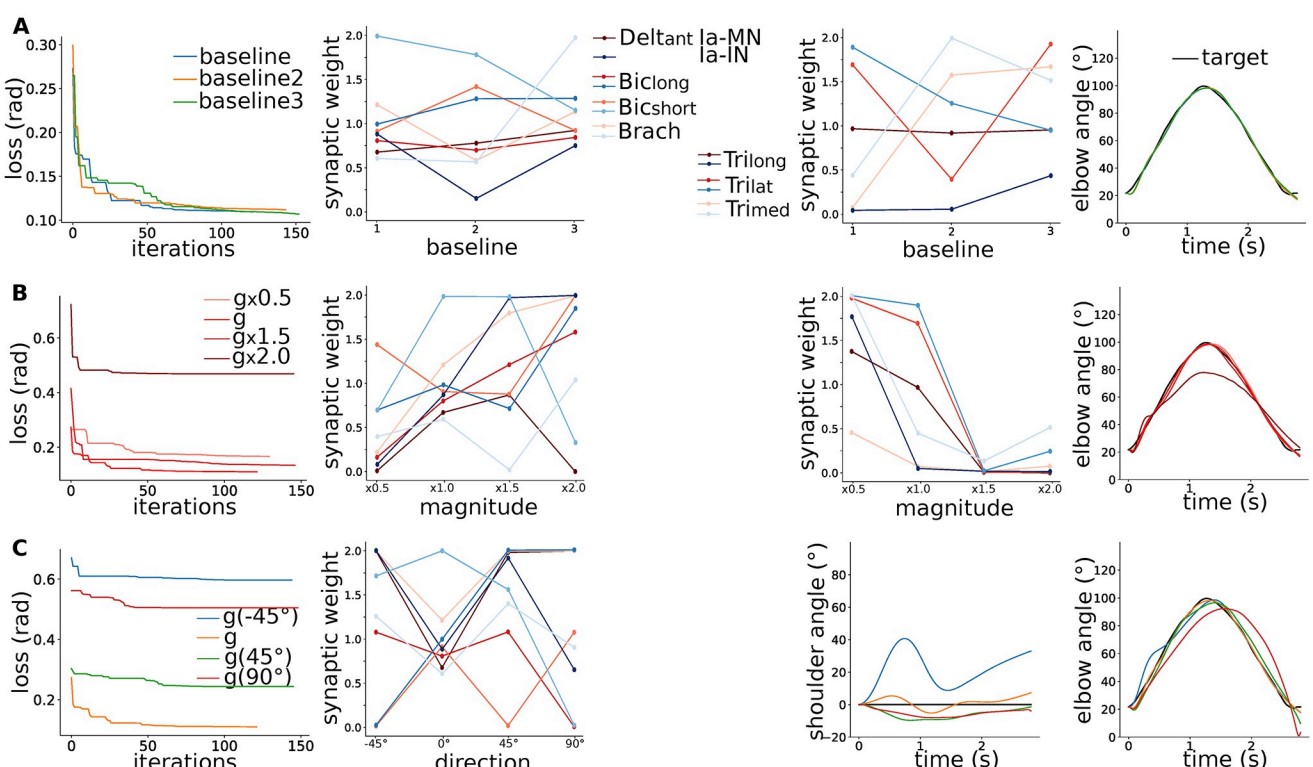

**Fig 5. Modulation of the spinal pathways in various gravity conditions.** A) Optimisation of the spinal synaptic strengths to reproduce the flexion-extension baseline: from left to right the resulting loss over optimisation iterations, the final flexor and extensor spinal synaptic strengths, and the resulting elbow trajectories compared to the target. **B)** Same for various gravity magnitudes. **C)** Same for various gravity directions.

## 3 Discussion

We implemented a modular spinal cord model receiving supraspinal sinusoidal commands and muscular sensory feedback to control a musculoskeletal upper limb model to reproduce elbow flexion-extension and circular recorded trajectories. These two movements were respectively 2.8 and 1.3 seconds long, with a range of motion of 78˚ and 40˚. The shoulder flexion-extension is also unrestricted, so that we consider a double joint control. This way we could check the behaviour of our model for various speeds and ROMs. The presence of SC facilitates the reproduction of the trajectories compared to a scenario without SC similarly to [27]. While the previous paper focused on how a simple SC interacts with the cerebellum to support motor learning and adapt to perturbations, the current paper explores a broader range of spinal circuits and their modulation in various scenarios.

We then showed that Ia-MN stretch reflex, Ia-MNs heteronymous connections and II stretch reflex lead to smoother movement with increasing synaptic strength as [13] argued for a role of these pathways in motor stability. Nevertheless, Ia-MNs heteronymous pathways and II stretch reflex strongly limit the range of motion. Ia-MN stretch reflex and Ia-MNs heteronymous connections also lead to robustness against perturbation as supported by [24, 25]. Nevertheless, the Ib autogenic inhibition pathway did not lead to robustness against perturbation, while [26] model highlighted such property. Sensitivity analyses confirmed that the stretch reflexes had the stronger effect on the movement smoothness and robustness against perturbation.

We finally considered a "minimal SC" including the Ia-MN stretch reflex and Ia-INa reciprocal inhibition similarly to [27]. We optimised on one hand the supraspinal commands, and on the other hand the synaptic strength in various gravity conditions. The main supraspinal commands modulation was found in the flexor sinusoid's amplitude, resp. frequency, to adapt to different gravity magnitudes, resp. directions. The modulation of the spinal synaptic strengths also remarkably reproduced the target flexion-extension trajectory for the mild gravity changes. The target circular trajectory could also be reproduced, even for the mild and more severe gravity changes. This trajectory indeed exhibit a smaller ROM that facilitates the severe gravity magnitude context, and a symmetry to gravity direction that facilitates the gravity direction context. We highlighted that both strategies of modulation of the supraspinal commands or of the spinal pathways can reproduce similar trajectories in different gravity environments. In both cases, we optimised motor control parameters within a combined control scheme of supraspinal and SC systems to adapt the model to different gravity conditions. However, as SC reflexes typically bring the musculoskeletal system back to its resting position (both physiologically and as defined in the model), it is not straightforward that modulating SC reflexes would allow for gravity compensation. To our knowledge, it is the first model studying and revealing spinal control mechanisms involved in gravity compensation.

Our model could still be improved in several directions. First of all, we considered supraspinal sinusoidal commands for simple elbow movements. We could design more sophisticated supraspinal commands for more complex movements. We provided supraspinal sinusoidal commands to the elbow flexor and extensor muscles only to simplify the optimisation problem, that could be extended by additional supraspinal commands to the shoulder flexor and extensor muscles. We could also consider more sophisticated supraspinal commands such as cerebellar learning signals [27]. The SC model also presents some limitations. More pathways could be explored, such as Renshaw cell connections to antagonist interneurons and Renshaw cells. The "minimal SC" could also be extended with additional pathways. We focused our study on synaptic strength, while other neuronal dynamics parameters could also be explored, especially the neuronal activation offset and sensory feedback delay. Additionally, we

optimised the supraspinal commands and spinal synaptic strength separately. In the human system, control and modulation are naturally distributed among the two components. While a combined optimisation accompanied by a sensitivity analysis could have provided insights into the primary control roles, we chose to separate the processes to simplify interpretation. This approach allowed us to emphasise that the spinal cord can independently handle gravity compensation.

Overall, the present results are deemed of great interest as they highlight that both strategies of modulation of the supraspinal commands or of the spinal stretch pathways can be used to control movements in different gravity environments. The spinal cord could thus assist gravity compensation. This study could then be extended by investigating the mechanisms of such modulation to understand further these control strategies.

## 4 Methods

### 4.1 Ethics

Experiments were approved by the Commission cantonale d'éthique de la recherche sur l'être humain du canton de Vaud (CER-VD) under the license number 2017-02112 and performed in accordance with the Declaration of Helsinki. The participant gave his written consent.

We implemented a modular SC model integrating supraspinal commands and direct muscle sensory feedbacks to control a upper limb musculoskeletal model to study various scenarios. The following subsections describe the different components of our integrated model, the scenarios of interest, and the metrics used to quantify results.

### 4.2 Musculoskeletal upper limb model

We simulated a two DOF musculoskeletal upper limb model adapted from [30]. The model included two flexion-extension joints, the shoulder and the elbow, actuated by seven Hill-based muscles [31]. The deltoid anterior ($Delt_{ant}$) and biceps long ($Bic_{long}$) are shoulder flexor muscles, whereas the triceps long ($Tri_{long}$) is a shoulder extensor. The biceps long and short ($Bic_{short}$) and the brachialis ($Brach$)) are elbow flexors, while the triceps long, lateral and medial ($Tri_{lat}$, $Tri_{med}$)) are elbow extensors. Note that $Bic_{long}$) and $Tri_{long}$) were biarticular muscles, as they actuated both joints. The antagonist and synergistic relation between muscles is depicted in Table 3 [6, 32]. The Hill-based muscle dynamics were the following:

$$\begin{cases} f_m = (a * f_{lv}(l_m, \dot{l}_m) + f_p(l_m)) * cos\theta \\ \dfrac{da}{dt} = \dfrac{u - a}{\tau(u, A)} \end{cases} \tag{1}$$

with $f_m$ the muscle force, $f_{lv}$ a combination of the force-length and force-velocity curves, $f_p$ the

**Table 3. Antagonist (x) and synergistic (o) relations between muscles.**

| Muscles | $Delt_{ant}$ | $Tri_{long}$ | $Tri_{lat}$ | $Tri_{med}$ | $Bic_{long}$ | $Bic_{short}$ | $Brach$ |
|---|---|---|---|---|---|---|---|
| $Delt_{ant}$ | - | x | | | o | | |
| $Tri_{long}$ | x | - | o | o | x | | |
| $Tri_{lat}$ | | o | - | o | | x | x |
| $Tri_{med}$ | | o | o | - | | x | x |
| $Bic_{long}$ | o | x | | | - | o | o |
| $Bic_{short}$ | | | x | x | o | - | o |
| $Brach$ | | | x | x | o | o | - |

passive force-length curve, $\theta$ the pennation angle, $a$ the muscle activation (i.e., the concentration of calcium ions within the muscle), and $u$ the muscle excitation (i.e., the firing of the MN) [31]. Both $u$ and $a$ are in [0, 1].

We used OpenSim physics engine to simulate the muscle and skeleton dynamics [29]. We scaled the OpenSim upper limb model to the participant's morphology to allow using kinematics and EMG from lab recordings. This scaling process was achieved using OpenSim scaling tool and OpenPifPaf Human Pose Estimation algorithm [33] during a static period.

## 4.3 Spinal cord model

To control these muscles, we built a modular SC model including for each muscle an exhaustive list of pathways that are commonly reported in the literature [6]. We considered the following pathways sketched on Fig 1C:

- Ia stretch reflex (Ia-MN): monosynaptic excitatory pathway from muscle stretch feedback conveyed by Ia fiber

- Reciprocal Ia inhibition between antagonist muscles (Ia-INa): disynaptic inhibitory pathway from antagonist stretch feedback conveyed by antagonist Ia fiber

- Ia heteronymous stretch reflex between synergistic muscles (Ia-MNs): monosynaptic excitatory pathway from synergistic stretch feedback conveyed by synergistic Ia fiber

- II static stretch reflex (II): disynaptic excitatory pathway from static stretch feedback conveyed by II fiber

- Ib autogenic inhibition (Ib): disynaptic inhibitory pathway from muscle tension feedback conveyed by Ib fiber

- Renshaw cell recurrent inhibition (RN): monosynaptic inhibition pathway from motoneuron activity conveyed by Renshaw cell. Other inhibitory connection are reported between Renshaw cells and antagonist Renshaw cells and Ia interneurons. We did not consider them to simplify the analysis.

The SC model thus integrated the supraspinal commands and the muscle sensory feedback (Fig 1A). For each muscle, a motoneuron (MN) received an excitatory connection conveying the supraspinal commands ($c_F$, resp. $c_E$, for flexor, resp. extensor muscle), and excitatory or inhibitory connections from the spinal pathways. For instance for Ia-MN stretch reflex, the MN receives an excitatory connection directly from the Ia afferent fibre of the muscle. Each neuron of these pathways was modeled with leaky dynamics as follow:

$$\tau \dot{r}(t) = -r(t) + \sigma(\sum_i w_i r_i(t - \delta_i)), \tag{2}$$

where $r$ stands for the neuron firing rate; $\tau = 1$ms the activation time constant (we considered fast-response large neurons as in [21]); $\sigma(x) = \frac{1}{1+e^{D(x-x_0)}}$ a steep sigmoid function with an offset $x_0$ of 0.5 and a scaling factor $D$ of 8 mimicking the on-off behaviour of neurons; $i$ the neuron input signals; $w_i$ the synaptic strength of the input connection; $r_i$ the input activity; and $\delta_i = 30$ms the sensory response delay in upper limbs [34, 35]. The synaptic strength of each pathway was evaluated between 0 and 1. For multi-synaptic pathways, we varied only the synaptic strength of the last connection to the MN and fixed the other ones to 1. For Ia-MNs heteronymous connections, the synaptic strength of each Ia-MN connection was divided by the number of connections received by the motoneuron to limit saturating input. In the optimisation scenario, the synaptic strengths were optimised between 0 and 2.

The sensory signals from the Ia, II and Ib fibers were modeled using Prochazka's fiber rate models [36], with a mean firing rate of 10Hz [21, 37, 38]:

$$\begin{cases} r_{Ia}(t) = sgn(\dot{l}_m(t)) * 4.3|\dot{l}_m(t)|_+^{0.6} + 2(l_m(t) - l_{0,m}) + 10 \\ r_{II}(t) = 13.5(l_m(t) - l_{0,m}) + 10 \\ r_{Ib}(t) = \dfrac{F_m(t)}{F_{max,m}} \end{cases} \tag{3}$$

where $l_m$ and $\dot{l}_m$ describes the muscle fibre length and velocity in mm and mm/s; $|x|_+ = max(|x|, 0.01)$; and $F_m(t)$ and $F_{max,m}$ are the muscle force and maximal isometric force. The output rates were scaled by their maximum to get a normalised value in [0, 1].

The MN output rates were finally provided as muscle excitation signals to the musculoskeletal model. We used FARMS Python library, developed at the BioRobotics laboratory [39] to model this modular SC.

## 4.4 Supraspinal commands

We defined supraspinal sinusoidal commands to reproduce various recorded elbow trajectories. These target trajectories are discrete like reaching and return tasks, and we considered one cycle of sinusoidal signals per elbow synergistic muscle group, namely the elbow flexor ($c_F$) and extensor ($c_E$) muscles, to simplify the optimisation problem, while representing the reaching movement and the return to the initial position. A motor control hypothesis indeed states that the motor system simplifies the production of movements by combining a small number of muscle synergies, or motor primitives. Several studies thus characterized the spatio-temporal organization of the phasic patterns of shoulder and arm muscles during reaching movements [40–42].

$$\begin{cases} c_F(t) = a_F sin(2\pi f_F t + \phi_F) + d_F \\ c_E(t) = a_E sin(2\pi f_E t + \phi_E) + d_E \end{cases} \tag{4}$$

where $a_F$ and $a_E$ describe the sinusoids' amplitudes; $f_F$ and $f_E$ the frequencies; $\phi_F$ and $\phi_E$ the phases; and $d_F$ and $d_E$ the offsets. Depending on the scenario, these parameters were optimised or fixed in the range of values summarised in Table 4. Note that the deltoid muscle does not receive any supraspinal command as it does not actuate the elbow joint.

## 4.5 Various scenarios

We first investigate the role of various spinal models in voluntary movement and robustness against perturbations. Then, we study the modulation of the supraspinal commands, and of the spinal pathways that showed the more notable effects in the previous scenarios in various gravity environments.

**Table 4. Supraspinal paramaters' range of value.**

| Parameter | Range |
|---|---|
| Amplitude $a$ | [0.05, 0.7] |
| Frequency $f$ | [0.1, 0.8] |
| Phase $\phi$ | [0, 2$\pi$] |
| Offset $d$ | [0, 0.1] |

**4.5.1 Behaviour of the integrated model for various recorded trajectories.** We first optimised the supraspinal commands without and with a "minimal SC" to reproduce a baseline recorded trajectory 2.8 seconds long for 78˚ range of motion (ROM) and a baseline circular trajectory of 1.3 seconds long and 40˚ of ROM. The shoulder flexion-extension is also unrestricted, so that we consider a double joint control. This way we could check the behaviour of our model for various speeds and ROMs. Our "minimal SC" comprised the Ia-MN stretch reflex and the Ia-INa reciprocal inhibition, this choice is motivated from scenarios S1 and S2 results. We repeated each optimisation three times to check the solution space and reproducibility.

The trajectory recordings involved a participant performing repetitive planar elbow flexion-extension and circular movements in the vertical plane in standing position at various self-selected speeds and range of motion. Prior to the experiments, approval was obtained from the Commission cantonale d'éthique de la recherche sur l'être humain du canton de Vaud (CER-VD) under license number 2017-02112. Written consent was obtained from then participant, and the experiments were conducted at NeuroRestore laboratory, Lausanne CHUV, in accordance with the principles outlined in the Declaration of Helsinki.

Kinematic data was captured using an RGB-D camera, and the OpenPifPaf human pose estimation algorithm [33] was used to extract the 2D positions of anatomical joints at a frame rate of 25fps. The 3D pose was then deduced from the camera depth information after correcting for distortion. Specially designed filters were also used to remove occlusions while ensuring consistency in joint anatomy and temporal coherence. We computed inverse kinematics (IK) to derive joint position and velocity from the joint positions. EMG recordings were conducted using the Delsys system and Trigno Avanti and Trigno Quattro sensors with an acquisition frequency of 1259.3Hz. All the muscles of the model were recorded, except the triceps medial that is a profound muscle. Alignment of EMG with kinematics signals was achieved using a trigger that induced a pulse in an additional EMG channel and illuminated an LED within the camera's frame.

**4.5.2 Scenario 1: Voluntary movements.** We optimised supraspinal commands without SC to reproduce recorded elbow trajectories. We then evaluated the effect of various spinal pathways in terms of movement smoothness and muscle recruitment with EMG. To do so, we considered each pathway individually and simulated the resulting trajectory with increasing synaptic strength, from 0 to 1.

**4.5.3 Scenario 2: Perturbed environment.** We performed a perturbation study by applying an additional force perturbations during the movement. A perturbation force of 30N was applied during 30ms to the hand, acting upward during the flexion-extension trajectory and downward during the circular trajectory, when the arm is flexed in both case. The robustness against perturbation of the various spinal pathways was evaluated by comparing the deviation to the position without perturbation. As the previous scenario, we considered each pathway individually and simulated the resulting trajectory with various pathway synaptic strength, from 0 to 1.

**4.5.4 Scenario 3: Modulation of the supraspinal commands in various gravity environments.** We considered a "minimal SC" model including the pathways that showed the more notable effects in the previous scenarios, namely the Ia-MN stretch reflex and Ia-INa reciprocal inhibition. Then, we optimised the supraspinal commands to reproduce the target trajectory in various gravity magnitude and direction conditions (magnitude from 0.5 to 2 times g, and direction rotation from -45˚ to 90˚ corresponding to the lying position).

**4.5.5 Scenario 4: Modulation of the spinal pathways in various gravity environments.** We finally considered the "minimal SC" model with modular synaptic strength, and the supraspinal commands optimised in S3 in normal gravity condition. We optimised the

synaptic strengths of the SC model between 0 and 2 in the same gravity conditions as above. With these various scenarios, we could investigate the role of various spinal pathways in voluntary movement and robustness against perturbation. We then study the modulation of supraspinal commands and spinal pathways in various gravity environments.

## 4.6 Optimisation framework

To optimize the supraspinal command parameters or the spinal synaptic strengths depending on the scenario, we use the Covariance Matrix Analysis Evolution Strategy (CMA-ES) [43]. We considered a loss function including the root mean square error (RMSE) between the simulated and the target trajectories, and a term penalising high muscle activation at the beginning and end of the movement as follow:

$$L = \sqrt{\frac{1}{N}\sum_{n=1}^{N}(\theta_{sh,s,n} - \theta_{sh,t,n})^2} + \sqrt{\frac{1}{N}\sum_{n=1}^{N}(\theta_{elb,s,n} - \theta_{elb,t,n})^2} + w_{pen}.\frac{1}{T}\sum_{n\in T}\frac{1}{M}\sum_{m=1}^{M}a_{m,t}^2 \qquad (5)$$

where $N$ stands for the number of time steps; $\theta_{sh,s}$ and $\theta_{elb,s}$ the simulated shoulder and elbow positions; and $\theta_{sh,t}$ and $\theta_{elb,t}$ the target shoulder and elbow positions. Regarding the penalty on the muscle activation, we considered the specific time frames of the simulation $T$ in the first and last 0.1s of the simulation. $M$ is the number of muscles, and the penalty weight $w_{pen}$ was set by trial and error to 5.

## 4.7 Performance metrics

**4.7.1 Measuring kinematics performance.** To evaluate the kinematic performance of the models, we defined the root mean square error (RMSE) between the simulated and the target trajectories as follow:

$$RMSE = \frac{1}{2}\left(\sqrt{\frac{1}{N}\sum_{n=1}^{N}(\theta_{sh,s,n} - \theta_{sh,t,n})^2} + \sqrt{\frac{1}{N}\sum_{n=1}^{N}(\theta_{elb,s,n} - \theta_{elb,t,n})^2}\right) \qquad (6)$$

where $N$ stands for the number of time steps; $\theta_{sh,s}$ and $\theta_{elb,s}$ the simulated shoulder and elbow positions; and $\theta_{sh,t}$ and $\theta_{elb,t}$ the target shoulder and elbow positions.

We evaluated the smoothness of the simulated movement by computing the elbow speed arc length (*SAL*, negative metrics, closer to 0 for smoother movements) [44, 45]:

$$SAL = -\sum_{n=1}^{N}\sqrt{\frac{1}{N}^2 + \dot{\theta}_{elb,s,n}^2} \qquad (7)$$

We also performed a sensitivity analysis on these two metrics to asses the effect of our 6 spinal reflexes. We used SALib python library to run a Sobol' sensitivity analysis [46]. We generated 7168 samples with a Saltelli sampler to compute second-order indices ($N^*(2^*D + 2)$ with $N = 512$ and $D = 6$ our 6 spinal reflex weight variables).

**4.7.2 Measuring robustness against perturbations.** To assess the robustness against perturbations, we computed the average deviation to the trajectory without perturbation:

$$dev = max(\theta_{elb,s,n}) - \theta_{elb,s,p-1} \qquad (8)$$

where $\theta_{elb,s,\,p-1}$ describes the elbow position one time step before the perturbation.

We also performed a sensitivity analysis on this metric as described above.

**4.7.3 Measuring physiological muscle activation.** We also evaluated physiological muscle activation by comparing the simulated muscle activation with recorded EMG. Simulated

activation signals are commonly compared to EMG envelopes. Nevertheless, such comparison is not straightforward due to scaling issues, as EMG signals are difficult to normalise and subject to measurement errors [47]. Additionally, our musculoskeletal model accounts for a subset of the numerous human upper limb muscles, which further challenges the direct comparison between recorded EMG and muscle activation signals. To overcome this issue, we adopted a more qualitative approach by comparing their global activation patterns.

We first computed the EMG envelopes by rectifying and low pass filtering the signals using a 5th order Butterworth filter with a cut-off frequency of 5Hz. We also recorded the maximal voluntary contraction (MVC) signals to normalise the EMG signals. We applied the same previous processing steps to the MVC signals, and normalised each EMG signal by the maximum of the corresponding MVC signal. To compare with the simulated activation, we finally interpolated the resulting EMG envelops on the corresponding simulation time points.

We then averaged the elbow flexor and extensor EMG envelops and simulated activation signals. We did not consider the simulated activation signal of the triceps medial as it was not recorded. To detect activation time windows, we set a percentage (*perc*) of the root mean square (RMS) value of each EMG signal as a threshold above which a muscle was considered activated [48]. The percentages set were based on visual analysis of the obtained EMG curves and on trial and error for each of the flexor and the extensor muscle groups [48]; 85% for flexors and 65% for extensors. We finally computed the overlap between the EMG and simulated activation time windows in percent. More precisely, we considered as positive value when the two signal windows overlapped, and negative value when just one window was present, and summed as follow:

$$
\begin{cases}
\varnothing_{gp} = \dfrac{N_{overlap,gp}}{N_{EMG,gp}} - \dfrac{N_{EMG-sim,gp}}{N_{tot}} - \dfrac{N_{sim-sEMG,gp}}{N_{tot}} \\[2ex]
\varnothing_{tot} = \dfrac{1}{2}\left(overlap_{flex} + overlap_{ext}\right)
\end{cases}
\tag{9}
$$

where $\varnothing_{gp}$ stands for the overlap metric for a certain group of muscles; $N_{overlap}$ the number of time points where the EMG and simulated activation time windows overlapped; $N_{EMG}$ the number of time points of the EMG activation time windows; $N_{EMG-sim}$ the number of time points where the EMG time window activation is present but not the simulated activation one ($N_{sim-EMG}$ inversely); and $N_{tot}$ the number of time points of the full signals.

## Acknowledgments

We gratefully thank the participant for his patience and willingness to collaborate in the recording sessions.

## Author Contributions

**Conceptualization:** Alice Bruel.

**Data curation:** Alice Bruel.

**Formal analysis:** Alice Bruel, Lina Bacha, Emma Boehly, Constance De Trogoff, Luca Represa.

**Funding acquisition:** Gregoire Courtine, Auke Ijspeert.

**Investigation:** Alice Bruel, Lina Bacha, Emma Boehly, Constance De Trogoff, Luca Represa.

**Methodology:** Alice Bruel.

**Software:** Alice Bruel.

**Supervision:** Gregoire Courtine, Auke Ijspeert.

**Visualization:** Alice Bruel, Lina Bacha, Emma Boehly, Constance De Trogoff.

**Writing – original draft:** Alice Bruel.

**Writing – review & editing:** Alice Bruel, Gregoire Courtine, Auke Ijspeert.

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
