## [Decision Letter · Decision Letter 0]

7 Oct 2024

Dear Mrs Bruel,

Thank you very much for submitting your manuscript "Role and modulation of various spinal pathways for human upper limb control in different gravity conditions" for consideration at PLOS Computational Biology.

As with all papers reviewed by the journal, your manuscript was reviewed by members of the editorial board and by an independent reviewer. We had the hardest time to find other available reviewers, and one of the handling editors became unavailable. And so it goes...

In light of the review (below this email), we would like to invite the resubmission of a significantly-revised version that takes into account the reviewer's comments.

Upon resubmission, we might seek for the fresh opinion of another reviewer. Maybe this is suboptimal but it's useless to waste more time now.

We cannot make any decision about publication until we have seen the revised manuscript and your response to the reviewers' comments. Your revised manuscript is also likely to be sent to reviewers for further evaluation.

Sincerely,

Daniele Marinazzo

Section Editor

PLOS Computational Biology

Reviewer's Responses to Questions

**Comments to the Authors:**

Reviewer #1: How the brain and spinal cord (SC) work together to implement motor adaptation, is an important question. Therefore I was quite interested to read this paper, and the work reported has been carried out with sufficient care. However, having read the paper, I am not sure what fundamental techniques or physiological principles I have learned from it. Using a neuromechanical model considering a Hill model of muscle-tendon mechanics and various reflexes including the monosynaptic stretch reflex and heteronymous reflexes, the paper (a) first evaluates which reflexes have most relevant effect in a single joint elbow movement up and down. It then (b) examines how the gravity involved in these movements can be learned by either the brain, i.e. through identification of a simple motor command's parameters, or through identification of the parameters of two most relevant reflexes. Part (a) is presented well, though it could have a systematic sensitivity analysis, that could be an interesting contribution. My problem is more with part (b) as I do not understand what is the contribution here. As both the brain and SC participate to the control and their parameters are combined, it is not surprising that either can be used to adapt the behaviour. In fact from an identification point of view one could consider the combined parameters and use techniques such as observability to understand what each can contribute to or whether some adaptation could not be provided by one of them. Also as only one task is considered it is difficult to make any conclusion about whether adaptation of the spinal reflexes would be sufficient to provide motor adaptation in general.

Another problem that I see is that (as mentioned in the text) some of the same authors just published a paper in this journal, with a cerebellar model and a simple SC model. I have looked at that paper and like it and see a clear message and contribution. In contrast, I have problem to identify a clear message or contribution in the present manuscript.

Some technical issues that are on the muscle model. The monotonic stiffness increase with activation/force is not considered, which is one of the most important characteristic of muscle mechanics, see e.g. Kirsch et al. (1994), IEEE Transactions on Biomedical Engineering. Also the Hill muscle model considers both the force length and force velocity relationships. However, is it important or valid here to consider the force-length relationship, as how it influences the force depends on the muscle and movement, see e.g. Murray et al. (2000), Journal of Biomechanics.

Understanding how the brain and spinal cord (SC) collaborate to achieve motor adaptation is a significant question in neuroscience. I was therefore intrigued to read this paper on this topic. However, after reading it, I am uncertain about the fundamental techniques or physiological principles that I have gained from it.

The paper uses a neuromechanical model incorporating a Hill-type muscle-tendon mechanism and several reflex pathways, including the monosynaptic stretch reflex and heteronymous reflexes. The work is divided into two parts: (a) it evaluates which reflexes have the most relevant effects on single-joint elbow movements (flexion and extension), and (b) it explores whether gravity's influence on these movements can be learned through either the brain's motor commands or by adapting the two most relevant reflexes.

I found that part (a) is well-presented, though it would benefit from a more systematic sensitivity analysis, which could enhance the contribution of this section. My primary concern lies with part (b), as the contribution here is unclear to me. Given that both the brain and SC are involved in motor control and their parameters are combined, it is unsurprising that either system could adapt behavior. From an identification perspective, one could consider the combined parameters and apply methods like observability to assess their individual roles in adaptation. In which case could only one of these components work, or can any of them work over a large repertoire of behaviours? Since only a single task is analyzed, it is difficult to generalize whether spinal reflex adaptation alone is sufficient for motor adaptation across a broader range of tasks.

Another issue is the overlap with a recent nice study by the same authors, which used a cerebellar model and a simplified SC model. I found that previous work to be clear in its message and contribution, whereas I struggled to identify a clear takeaway from the current manuscript.

There are also some technical concerns regarding the muscle model. The monotonic increase in stiffness with activation/force, which is a key characteristic of muscle mechanics (see Kirsch et al. 1994, IEEE Transactions on Biomedical Engineering), is not addressed. Additionally, while the Hill model considers both force-length and force-velocity relationships (and the force velocity is clearly necessary), it is unclear if it is necessary or valid to include the force-length relationship in this context, as its influence varies depending on the specific muscle and movement (see Murray et al. 2000, Journal of Biomechanics).

**Have the authors made all data and (if applicable) computational code underlying the findings in their manuscript fully available?**

Reviewer #1: None

PLOS authors have the option to publish the peer review history of their article (what does this mean?). If published, this will include your full peer review and any attached files.

Reviewer #1: No
---

## [Editor Report · Decision Letter 1]

18 Dec 2024

Dear Mrs Bruel,

We are pleased to inform you that your manuscript 'Role and modulation of various spinal pathways for human upper limb control in different gravity conditions' has been provisionally accepted for publication in PLOS Computational Biology.

Best regards,

Daniele Marinazzo

Section Editor

PLOS Computational Biology

Daniele Marinazzo

Section Editor

PLOS Computational Biology

---

## [Editor Report · Acceptance letter]

27 Dec 2024

PCOMPBIOL-D-24-00614R1 

Role and modulation of various spinal pathways for human upper limb control in different gravity conditions

Dear Dr Bruel,

I am pleased to inform you that your manuscript has been formally accepted for publication in PLOS Computational Biology. Your manuscript is now with our production department and you will be notified of the publication date in due course.

With kind regards,

Zsofia Freund
